

# Image based visual servoing with kinematic singularity avoidance for mobile manipulator

Jesus Hernandez-Barragan*, Carlos Villaseñor, Carlos Lopez-Franco, Nancy Arana-Daniel and Javier Gomez-Avila*

University Center for Exact Sciences and Engineering, Universidad de Guadalajara, Guadalajara, Jalisco, Mexico
* These authors contributed equally to this work.

## ABSTRACT

This article presents an implementation of visual servoing (VS) for a redundant mobile manipulator in an eye-in-hand configuration. We used the image based visual servoing (IBVS) scheme, which means the pose control of the robot is based on the error features in the image of a camera. Conventional eye-in-hand VS requires the inversion of a Jacobian matrix, which can become rank deficient, provoking kinematic singularities. In this work, the inversion of the Jacobian matrix is solved using damped least squares (DLS) to reduce singularities and smooth out discontinuities. In addition, a task prioritization scheme is proposed where a primary task performs the eye-in-hand IBVS task, and a secondary task maximizes a manipulability measure to avoid singularities. Finally, a gravity compensation term is also considered and defined on the basis of the image space error. The effectiveness of the proposed algorithm is demonstrated through both simulation and experimental results considering the Kuka YouBot.

## INTRODUCTION

Visual servo control refers to the use of computer vision data control the pose of a robot (*Chaumette & Hutchinson, 2006, 2007; Wang et al., 2024; Leomanni et al., 2024; Kumar et al., 2024*). Estimating the pose of a robot is a crucial task in robotics (*Ban et al., 2024*). It forms the basis for numerous subsequent tasks, such as visually-guided object grasping and manipulation (*Li et al., 2023; Tian et al., 2023; Zuo et al., 2019*), autonomous navigation (*Bultmann, Memmesheimer & Behnke, 2023*), multi-robot collaboration (*Li, De Wagter & De Croon, 2022; Papadimitriou, Mansouri & Nikolakopoulos, 2022*) and human-robot interaction (*Christen et al., 2023*).

Exteroceptive sensors, such as cameras, offer a robust alternative that relies on external references rather than internal measurements. That is the reason why image based visual servoing (IBVS) control schemes have become so popular. In IBVS, the primary objective is to minimize the error between the current image features captured by a camera and the desired image features. This error is computed in the image space, and the control laws are designed to drive this error to zero, thereby achieving the desired pose of the robot. The camera used in IBVS can be configured in two ways: eye-to-hand, where the camera is

Corresponding author
Javier Gomez-Avila,
jenrique.gomez@academicos.udg.mx

fixed in the environment, or eye-in-hand, where the camera is mounted on the end-effector, which is the approach used in this work (*Flandin, Chaumette & Marchand, 2000*).

In this context, the implementation of IBVS is particularly beneficial for redundant manipulators, which require advanced control strategies to exploit their additional degrees of freedom effectively. Through the combination of visual feedback and advanced control terms, IBVS enables these robots to perform complex tasks with high precision and reliability. Some related works include (*Ren, Li & Li, 2020*; *Heshmati-Alamdari et al., 2014*; *Karras et al., 2022*; *Heshmati-Alamdari et al., 2024*), where IBVS is implemented in redundant manipulators using feature constraints to keep image features in the field of view.

Several studies have explored using redundancy to define different types of constraints by incorporating secondary tasks (increase manipulability) alongside the main task (visual servoing) (*Mansard & Chaumette, 2009*; *Yoshikawa, 1996*; *Chaumette & Marchand, 2001*; *Rastegarpanah, Aflakian & Stolkin, 2021*). However, these works do not consider mobile platforms.

In recent years, the introduction of mobile manipulators into the industry has transformed the viewpoint of both users and developers. These robots enhance the manipulation abilities of articulated industrial arms by extending their reach across the entire workshop area. However, in current industrial applications, the navigation and manipulation phases of mobile manipulators are typically separated for technical simplicity and safety considerations (*Marvel & Bostelman, 2013*; *Markis et al., 2019*); Nevertheless, it is possible to achieve cooperation between the mobile platform and manipulator arm simultaneously with visual servoing approaches (*González Huarte & Ibarguren, 2024*; *Wang, Lang & De Silva, 2009*; *Lippiello, Siciliano & Villani, 2007*).

Recently, hybrid approaches that combine IBVS and position-based visual servoing (PBVS) have been explored (*Li & Xiong, 2021*; *Jo & Chwa, 2023*). These approaches offer increased robustness to feature occlusion and compensate for inaccurate pose estimation due to slippage errors in odometry and wheel encoders, as IBVS does not rely on the internal odometry of the robot but instead uses visual feedback from the environment. However, hybrid visual servoing requires the integration of both image-based and position-based control loops, which increases the system's complexity, making the design, tuning, and implementation of the control system more challenging.

Most research on visual servoing and mobile robots focuses on mobile robot navigation, with only a few instances of vision-based mobile manipulation being documented (*Wang, Lang & De Silva, 2009*). Many of the works that address the visual control of mobile platforms include non-holonomic constraints (*Silveira et al., 2001*; *De, Bayer & Koditschek, 2014*; *Huang & Su, 2019*; *Zhao & Wang, 2020*; *Dirik, Kocamaz & Dönmez, 2020*; *Li & Xiong, 2021*; *Yu & Wu, 2024*). It should be emphasized that when a mobile platform has non-holonomic constraints, it reduces the possibility of exploiting kinematic redundancies.

This work proposes a VS scheme for an eye-in-hand IBVS task on a redundant omnidirectional mobile manipulator. In addition, the proposed scheme deals with singularities based on DLS and manipulability measure maximization. To reduce

singularities and smooth out the discontinuities, the inversion of the Jacobian matrix is solved using DLS. Since the mobile manipulator is redundant, a task prioritization scheme is proposed to perform the eye-in-hand IBVS as a primary task while maximizing a manipulability measure as a secondary task. Finally, a gravity compensation term is considered, especially for the case of real-world applications. The effectiveness of the proposed algorithm is demonstrated through both simulation and experimental results considering the Kuka YouBot which is an 8-DOF omnidirectional mobile manipulator. The simulation codes are shared to facilitate the replicability of this work.

The rest of this research is organized as follows: In "Image Based Visual Servoing (IBVS)" the reader will find an overview of visual servoing. In "Eye-in-hand IBVS for omnidirectional mobile manipulators", the proposed eye-in-hand IBVS approach is addressed. This scheme is tested in both simulation and real-world experiments, the results are presented in "Experimental Results". Then, An analysis of the obtained results and the future research directions are given in "Discussion". Finally, the conclusions are discussed in "Conclusions".

## IMAGE BASED VISUAL SERVOING

Multiple schemes of visual servoing are found in the literature *Chaumette & Hutchinson (2006, 2007)*. In this work, we implement IBVS with an eye-in-hand configuration. This approach is described below.

This method involves extracting relevant information from the visual input to guide the actions of the robots, such as positioning, navigating, or manipulating objects. The primary objective is to minimize the error between the current visual perception and the desired visual state

$$\mathbf{e}(t) = \mathbf{s}(\mathbf{m}(t), \mathbf{a}) - \mathbf{s}^* \tag{1}$$

where $s$ is the vector of coordinates in the image plane, $\mathbf{m}(t) = (u, v)$ are the measurements of interest points (in pixel units) and $\mathbf{a} = (c_u, c_v, f, \alpha)$ are additional parameters of the system (in this case, the intrinsic parameters of the camera): $c_u$ and $c_v$ are the coordinates of the principal point, $f$ the focal length and $\alpha$ the ratio of the pixel dimensions. $\mathbf{s}^*$ is the vector of the desired coordinates.

The next step is to calculate a velocity controller based on the relationship between the time variation of $\mathbf{s}$ and the velocity of the camera, considering the case of controlling it with six degrees of freedom (DOF).

$$\mathbf{v}_c = (v_c, \omega_c) \tag{2}$$

where $v_c$ and $\omega_c$ are the instantaneous linear and angular velocity of the origin of the camera. Features velocity $\dot{\mathbf{s}}$ and $\mathbf{v}_c$ are related by a matrix known as interaction matrix and denoted by $\mathbf{L}_s$

$$\dot{\mathbf{s}} = \mathbf{L_s} \mathbf{v}_c \tag{3}$$

where $\mathbf{L_s} \in \mathbb{R}^{k \times 6}$ and $k$ is $2\times$ the number of features. For each feature $i$, the interaction matrix is defined as

$$\mathbf{L_s} = \begin{bmatrix} \mathbf{L_{s_1}} \\ \mathbf{L_{s_2}} \\ \vdots \\ \mathbf{L_{s_n}} \end{bmatrix} \tag{4}$$

where

$$\mathbf{L_{s_i}} = \begin{bmatrix} \frac{-1}{Z} & 0 & \frac{x}{Z} & xy & -(1+x^2) & y \\ 0 & \frac{-1}{Z} & \frac{y}{Z} & 1+y^2 & -xy & -x \end{bmatrix} \tag{5}$$

where $Z$ is the depth from the camera to the feature. In this work, this depth is estimated using prior knowledge of the ArUco pattern. $x$ and $y$ are the feature coordinates in the image plane given by

$$x = (u - c_u)/f\alpha \tag{6}$$
$$y = (v - c_v)/f \tag{7}$$

The relationship between camera velocity and the variation of the error is given by

$$\dot{\mathbf{e}} = \mathbf{L_s}\mathbf{v}_c \tag{8}$$

and, to guarantee an exponential reduction of the error, the camera velocities are calculated with

$$\mathbf{v}_c = -\lambda \mathbf{L_s}^+ \mathbf{e} \tag{9}$$

## EYE-IN-HAND IBVS FOR OMNIDIRECTIONAL MOBILE MANIPULATORS

This section provides the kinematic model for an omnidirectional mobile manipulator. Commonly, mobile manipulators are redundant robots which is beneficial for working with task priority schemes. The robot, with eight degrees of freedom, consists of an omnidirectional mobile platform with four mecanum wheels, allowing it to move to any position with a desired orientation (*Zhang et al., 2016*; *Wu et al., 2017*; *Kundu et al., 2017*).

### Omnidirectional mobile manipulator kinematics

The eye-in-hand system with coordinate frame assignments for robot kinematics is illustrated in Fig. 1. We start defining the following homogeneous matrices: ${}^w\mathbf{T}_b$ represents the position and orientation (pose) of the mobile platform, with respect to the world frame, ${}^b\mathbf{T}_m$ is a constant transformation between the mobile platform and the manipulator base, ${}^m\mathbf{T}_e$ is the transformation from the manipulator base to the end-effector frame. Finally, ${}^c\mathbf{T}_e$ is the position and orientation of the end-effector, with respect to the camera frame.

The mobile platform pose ${}^w\mathbf{T}_b$ is defined as

$${}^w\mathbf{T}_b(\mathbf{q}_b) = \begin{bmatrix} \cos(\theta_b) & -\sin(\theta_b) & 0 & x_b \\ \sin(\theta_b) & \cos(\theta_b) & 0 & y_b \\ 0 & 0 & 1 & 0 \\ 0 & 0 & 0 & 1 \end{bmatrix} \tag{10}$$

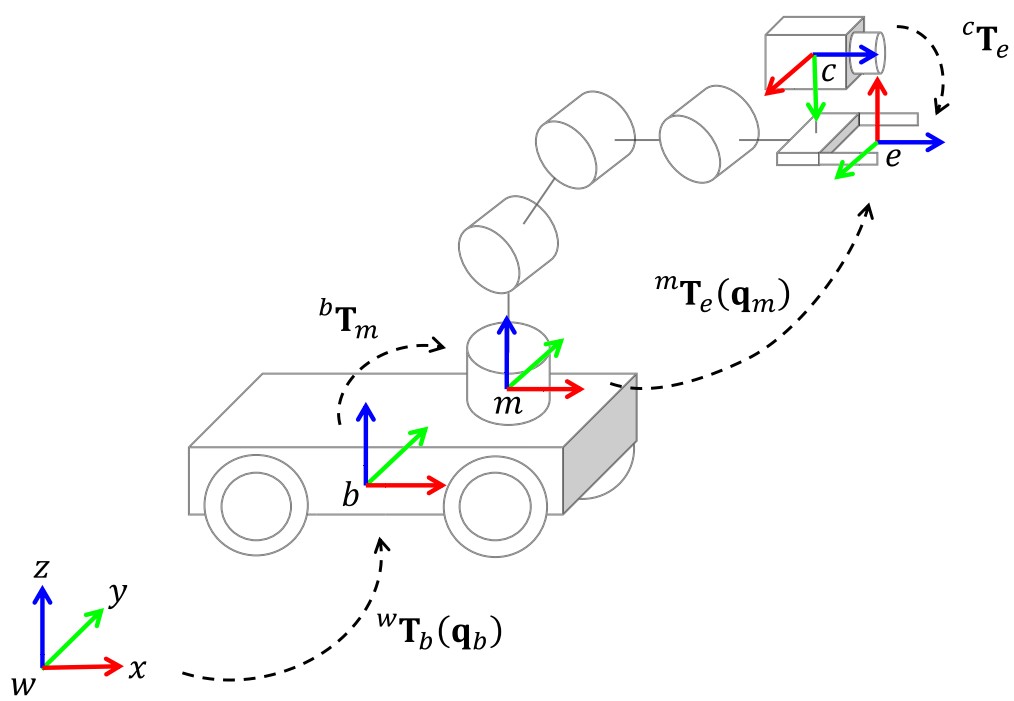

**Figure 1 Eye-in-hand system and coordinate frames assignment for a mobile manipulator.**

where $\mathbf{q_b} = \begin{bmatrix} x_b & y_b & \theta_b \end{bmatrix}^T$ represents the position $(x_b, y_b)$ and orientation $\theta_b$ of the omnidirectional platform.

Let $\mathbf{q_m} = \begin{bmatrix} q_1 & q_2 \cdots q_n \end{bmatrix}^T$ be the manipulator configuration with $n$ DOF. The manipulator kinematics ${}^m\mathbf{T}_e(\mathbf{q}_m)$ can be computed based on the Denavit-Hartenberg (DH) convention. Then, forward kinematic ${}^0\mathbf{T}_n(\mathbf{q}_m)$ can be defined as

$$
{}^m\mathbf{T}_e(\mathbf{q}_m) = {}^0\mathbf{T}_n(\mathbf{q}_m) = {}^0\mathbf{T}_1(q_1){}^1\mathbf{T}_2(q_2) \cdots {}^{n-1}\mathbf{T}_n(q_n) \tag{11}
$$

where ${}^{i-1}\mathbf{T}_i$ transforms the frame attached to the link $i-1$ into the frame link $i$ of the robot manipulator, with $i = 0, 1, 2, \cdots, n$.

The joint variable for the mobile manipulator is given by $\mathbf{q} = \begin{bmatrix} \mathbf{q}_b^T & \mathbf{q}_m^T \end{bmatrix}$. The forward kinematics ${}^w\mathbf{T}_e(\mathbf{q})$ of the mobile manipulator is obtained with

$$
{}^w\mathbf{T}_e(\mathbf{q}) = {}^w\mathbf{T}_b(\mathbf{q}_b){}^b\mathbf{T}_m{}^m\mathbf{T}_e(\mathbf{q}_m) = \begin{bmatrix} {}^w\mathbf{R}_e & {}^w\mathbf{t}_e \\ 0 & 1 \end{bmatrix} \tag{12}
$$

where ${}^w\mathbf{R}_e$ is the orientation and ${}^w\mathbf{t}_e$ the position of the end-effector, with respect to the world frame $w$.

Inverse kinematics involves calculating the joint variables $\mathbf{q}$ given the desired end-effector pose. This calculation can be achieved by minimizing an error function through an iterative process based on differential kinematics (*Sciavicco & Siciliano, 2012*). Differential

kinematics seeks to establish the relationship between the joint velocities $\dot{\mathbf{q}}$ and the end-effector velocity $\dot{\mathbf{x}}$.

$$\dot{\mathbf{x}} = \mathbf{J}(\mathbf{q})\dot{\mathbf{q}} \tag{13}$$

where $\mathbf{J}$ is a Jacobian matrix and $\dot{\mathbf{x}} = (v_e, \boldsymbol{\omega}_e)$. $v_e$ and $\boldsymbol{\omega}_e$ are instantaneous linear and angular velocities of the end-effector. Moreover, $\dot{\mathbf{q}} = \begin{bmatrix} \dot{\mathbf{q}}_b^T & \dot{\mathbf{q}}_m^T \end{bmatrix}$ where $\dot{\mathbf{q}}_b^T$ are the mobile platform velocities $(\dot{x}_b, \dot{y}_b, \dot{\theta}_b)$, and $\dot{\mathbf{q}}_m^T$ are the manipulator velocities in joint space.

The Jacobian matrix $\mathbf{J}$ can be computed as follows

$$\mathbf{J}(\mathbf{q}) = \begin{bmatrix} \mathbf{J}_v(\mathbf{q}) \\ \mathbf{J}_\omega(\mathbf{q}) \end{bmatrix} \tag{14}$$

where $\mathbf{J}_v(\mathbf{q})$ matrix relating the contribution of the end-effector linear velocity, and $\mathbf{J}_\omega(\mathbf{q})$ matrix relating the contribution of the end-effector angular velocity.

The matrix $\mathbf{J}_v(\mathbf{q})$ is defined as

$$\mathbf{J}_v(\mathbf{q}) = \begin{bmatrix} \frac{\partial t_x}{\partial x_b} & \frac{\partial t_x}{\partial y_b} & \frac{\partial t_x}{\partial \theta_b} & \frac{\partial t_x}{\partial q_1} & \frac{\partial t_x}{\partial q_2} & \cdots & \frac{\partial t_x}{\partial q_n} \\ \frac{\partial t_y}{\partial x_b} & \frac{\partial t_y}{\partial y_b} & \frac{\partial t_y}{\partial \theta_b} & \frac{\partial t_y}{\partial q_1} & \frac{\partial t_y}{\partial q_2} & \cdots & \frac{\partial t_y}{\partial q_n} \\ \frac{\partial t_z}{\partial x_b} & \frac{\partial t_z}{\partial y_b} & \frac{\partial t_z}{\partial \theta_b} & \frac{\partial t_z}{\partial q_1} & \frac{\partial t_z}{\partial q_2} & \cdots & \frac{\partial t_z}{\partial q_n} \end{bmatrix} \tag{15}$$

where ${}^w\mathbf{t}_e = \begin{bmatrix} t_x & t_y & t_z \end{bmatrix}^T$ is the end-effector position related to the joint variable $\mathbf{q}$. Moreover, the matrix $\mathbf{J}_\omega(\mathbf{q})$ is defined as

$$\mathbf{J}_\omega(\mathbf{q}) = \begin{bmatrix} 0 & 0 & {}^w\mathbf{z}_0 & {}^w\mathbf{z}_1 \cdots {}^w\mathbf{z}_{n-1} \end{bmatrix} \tag{16}$$

where ${}^w\mathbf{z}_m$ and ${}^w\mathbf{z}_i$ are the z-axis of the rotation matrices ${}^w\mathbf{R}_m$ and ${}^w\mathbf{R}_i$, respectively. Notice that the first two rows are set to $\mathbf{0}$ since $x_b$ and $y_b$ displacements do not provoke any end-effector angular velocity.

Rotation matrix ${}^w\mathbf{R}_i$ can be found in the homogeneous transformation ${}^w\mathbf{T}_i(\mathbf{q})$, which is

$$ {}^w\mathbf{T}_i(\mathbf{q}) = {}^w\mathbf{T}_b(\mathbf{q}_b) {}^b\mathbf{T}_m \prod_{j=1}^{j=i} {}^{j-1}\mathbf{T}_j(q_j) = \begin{bmatrix} {}^w\mathbf{R}_i & {}^w\mathbf{t}_i \\ \mathbf{0} & 1 \end{bmatrix}. \tag{17}$$

Finally, the mapping between the mobile platform $\dot{\mathbf{q}}_b^T$ to each wheel velocity is given by the following relationship

$$\begin{bmatrix} v_1 \\ v_2 \\ v_3 \\ v_4 \end{bmatrix} = \begin{bmatrix} \sqrt{2}\sin\left(\theta_b + \frac{\pi}{4}\right) & -\sqrt{2}\cos\left(\theta_b + \frac{\pi}{4}\right) & -(L+l) \\ \sqrt{2}\cos\left(\theta_b + \frac{\pi}{4}\right) & \sqrt{2}\sin\left(\theta_b + \frac{\pi}{4}\right) & (L+l) \\ \sqrt{2}\cos\left(\theta_b + \frac{\pi}{4}\right) & \sqrt{2}\sin\left(\theta_b + \frac{\pi}{4}\right) & -(L+l) \\ \sqrt{2}\sin\left(\theta_b + \frac{\pi}{4}\right) & -\sqrt{2}\cos\left(\theta_b + \frac{\pi}{4}\right) & (L+l) \end{bmatrix} \begin{bmatrix} \dot{x}_b \\ \dot{y}_b \\ \dot{\theta}_b \end{bmatrix} \tag{18}$$

where $v_i$ is the linear velocity of wheel $i$, $L$ is half of the distance between the front and the rear wheels, and $l$ is half of the distance between the left and the right wheels, see Fig. 2.

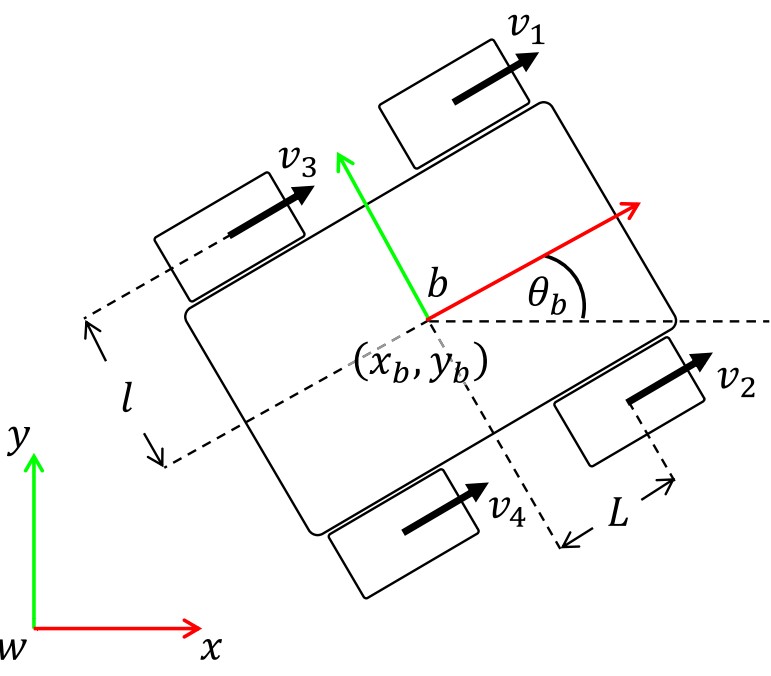

**Figure 2 Omnidireccional mobile platform consisting of four mecanum wheels.**

## Eye-in-hand scheme with task priority and gravity compensation

To consider an eye-in-hand scheme for mobile manipulator visual servoing, we propose to rewrite (Eq. (9)) to map the camera velocity control to the end-effector velocity expressed in the world frame, which is

$$\mathbf{v}_w = -\lambda(\mathbf{L_s}^c\mathbf{V}_w)^+\mathbf{e} \tag{19}$$

where $^c\mathbf{V}_w$ is a spatial motion transform matrix from the camera frame to the world frame, and $\mathbf{v}_w = (v_w, \omega_w)$ contains the linear $v_w$ and angular $\omega_w$ velocity expressed in the world frame. The matrix $^c\mathbf{V}_w$ is given as

$$^c\mathbf{V}_w = \begin{bmatrix} ^c\mathbf{R}_w & [^c\mathbf{t}_w]_\times \\ \mathbf{0} & ^c\mathbf{R}_w \end{bmatrix} \tag{20}$$

where $[^c\mathbf{t}_w]_\times$ is the skew-symmetric matrix associated with the vector $^c\mathbf{t}_w$, and $^c\mathbf{T}_w(^c\mathbf{R}_w,{}^c\mathbf{t}_w)$ is the transformation matrix from camera to world frame, which is computed as

$$^c\mathbf{T}_w(\mathbf{q}) = {}^c\mathbf{T}_e{}^w\mathbf{T}_e(\mathbf{q})^{-1} = \begin{bmatrix} ^c\mathbf{R}_w & ^c\mathbf{t}_w \\ \mathbf{0} & 1 \end{bmatrix}. \tag{21}$$

To map Cartesian velocities $\mathbf{v}_w = \dot{\mathbf{x}}$ to joint velocities $\dot{\mathbf{q}}$, using Eq. (13) we have

$$\dot{\mathbf{q}} = \mathbf{J}(\mathbf{q})^+\mathbf{v}_w \tag{22}$$

where $\mathbf{J}(\mathbf{q})^+$ is the Moore-Penrose pseudoinverse of $\mathbf{J}(\mathbf{q})$.

To overcome the problem of inverting $\mathbf{J}(\mathbf{q})^{+}$ in the neighborhood of a singularity is provided by the damped least-square (DLS) inverse

$$\mathbf{J}(\mathbf{q})^{+} = \mathbf{J}(\mathbf{q})^{T}\left(\mathbf{J}(\mathbf{q})\mathbf{J}(\mathbf{q})^{T} + \beta^{2}\mathbf{I}\right)^{-1} \qquad (23)$$

where $\beta$ is a damping factor that improves the conditioning of the inversion from a numerical viewpoint. DLS helps reduce the impact of singular configurations, which occur when $\mathbf{J}\mathbf{J}^{T}$ is not full rank (that is, when its determinant is zero or any of its eigenvalues are zero).

Consider the eigenvalue decomposition $\mathbf{J}\mathbf{J}^{T} = \mathbf{U}\Sigma\mathbf{V}^{T}$, the matrix $\Sigma$ is a diagonal matrix containing the eigenvalues $\{\sigma_{1}, \sigma_{2}, \ldots, \sigma_{n}\}$. If we invert the decomposition, we get $(\mathbf{J}\mathbf{J}^{T})^{-1} = \mathbf{V}\Sigma^{+}\mathbf{U}^{T}$, where $\Sigma_{ii}^{+} = \frac{1}{\sigma_{i}}$.

A singularity configuration will have some $\sigma_{i} = 0$; we get a stable inversion if we add a positive scalar $\beta^{2}$, as shown in Eq. (24).

$$\Sigma_{ii}^{+} = \frac{1}{\sigma_{i} + \beta^{2}}. \qquad (24)$$

Then, the stabilized $i$-eigenvalue will be $\sigma_{i} + \beta^{2}$. The final form (Eq. (25)) is the well-known solution of the Thikonov regularization (*Nair, Hegland & Anderssen, 1997*; *Gerth, 2021*).

$$\mathbf{U}(\Sigma + \beta^{2}\mathbf{I})\mathbf{V}^{T} = \mathbf{U}\Sigma\mathbf{V}^{T} + \beta^{2}\mathbf{U}\mathbf{V}^{T} = \mathbf{J}\mathbf{J}^{T} + \beta^{2}\mathbf{I}. \qquad (25)$$

Moreover, this work considers a task priority scheme to deal with feature error convergence of the visual servoing task while maximizing the mobile robot manipulability. The task priority scheme is given by

$$\dot{\mathbf{q}} = \mathbf{J}(\mathbf{q})^{+}\mathbf{v}_{w} + \left(\mathbf{I} - \mathbf{J}(\mathbf{q})^{+}\mathbf{J}(\mathbf{q})\right)\dot{\mathbf{q}}_{0} \qquad (26)$$

where $\mathbf{v}_{w}$ is the main task, and $\left(\mathbf{I} - \mathbf{J}(\mathbf{q})^{+}\mathbf{J}(\mathbf{q})\right)$ allows the projection of the second task $\dot{\mathbf{q}}_{0}$ into the null space of $\mathbf{J}$.

In this work, we propose a vector $\dot{\mathbf{q}}_{0}$ to avoid singularities based on the manipulability measure $m(\mathbf{q})$ defined as

$$m(\mathbf{q}) = \sqrt{\det\left(\mathbf{J}(\mathbf{q})\mathbf{J}(\mathbf{q})^{T}\right)}. \qquad (27)$$

The second task $\dot{\mathbf{q}}_{0}$ is calculated with

$$\dot{\mathbf{q}}_{0} = k_{0}\left(\frac{\partial m(\mathbf{q})}{\partial \mathbf{q}}\right) \qquad (28)$$

where $k_{0} > 0$. The robot redundancy is exploited to avoid kinematic singularities by maximizing the manipulability measure. Indeed, $\dot{\mathbf{q}}_{0}$ can be designed for collision avoidance by letting the joints reconfigure to avoid obstacles without affecting the primary task. Moreover, $\dot{\mathbf{q}}_{0}$ can also be designed for joint limit avoidance by guiding the joint

configuration towards the middle of their range while simultaneously performing the primary task.

Finally, a gravity compensation term is also considered, which is crucial for real-world implementations. Let us consider the gravitational potential energy as follows

$$\mathbf{u}(\mathbf{q}) = g \sum_{i=1}^{n} m_i h_i(\mathbf{q}) \tag{29}$$

where $g$ is the gravity acceleration, $m_i$ is the mass of the link $i$ and $h_i$ is the height of the center of mass and it depends on $\mathbf{q}$. For simplicity, we estimate the center of mass at the middle of the link.

The gravity vector $\mathbf{g}(\mathbf{q})$ is calculated with

$$\mathbf{g}(\mathbf{q}) = \frac{\partial}{\partial \mathbf{q}} \mathbf{u}(\mathbf{q}). \tag{30}$$

Finally, the term $\mathbf{g}(\mathbf{q})$ in Eq. (30) is added to Eq. (26)

$$\dot{\mathbf{q}} = \mathbf{J}(\mathbf{q})^+ \mathbf{v}_w + \left(\mathbf{I} - \mathbf{J}(\mathbf{q})^+ \mathbf{J}(\mathbf{q})\right) \dot{\mathbf{q}}_0 + \mathbf{g}(\mathbf{q}). \tag{31}$$

Equation (31) defines the proposed eye-in-hand scheme with task priority and gravity compensation for a mobile manipulator with an omnidirectional platform.

## EXPERIMENTAL RESULTS

This scheme is tested in both simulation and real-world experiments. The applicability is demonstrated on the Kuka YouBot which is an omnidirectional mobile manipulator.

Experiments are performed in three parts. First, an ideal simulation is performed where no gravity is required. Second, a simulation is performed in the Coppelia simulator. In this scenario, the proposed approach is tested under non-modeled dynamics and gravity compensation is required. Third, real-world experiments were performed on the Kuka YouBot robot with a conventional RGB camera.

The YouBot consists of a 5-DOF manipulator and a 3-DOF omnidirectional mobile platform. In terms of the mobile manipulator kinematics, the transformation $^w\mathbf{T}_b$ can be calculated using the pose of the mobile platform, defined by $(x_b, y_b, \theta_b)$ as shown in Eq. (10). The constant transformation $^b\mathbf{T}_m$ is represented as follows

$$^b\mathbf{T}_m = \begin{bmatrix} 1 & 0 & 0 & 0.140 \\ 0 & 1 & 0 & 0 \\ 0 & 0 & 1 & 0.151 \\ 0 & 0 & 0 & 1 \end{bmatrix}. \tag{32}$$

The transformation $^m\mathbf{T}_e = {}^0\mathbf{T}_5$ is obtained from the Denavit-Hartenberg table shown in Table 1. The joint variable $\mathbf{q}$ for the mobile manipulator will be

$$\mathbf{q} = \begin{bmatrix} x_b & y_b & \theta_b & \theta_1 & \theta_2 & \theta_3 & \theta_4 & \theta_5 \end{bmatrix}^T \tag{33}$$

where $\theta_1$ to $\theta_5$ represent the joint configuration of the manipulator.

**Table 1  DH table for the Kuka YouBot manipulator.**

| Joint | a (mm) | $\alpha$ (rad) | d (mm) | $\theta$ (rad) |
|---|---|---|---|---|
| 1 | 33 | $\pi/2$ | 147 | $\theta_1$ |
| 2 | 155 | 0 | 0 | $\theta_2$ |
| 3 | 135 | 0 | 0 | $\theta_3$ |
| 4 | 0 | $\pi/2$ | 0 | $\theta_4$ |
| 5 | 0 | 0 | 217.5 | $\theta_5$ |

The constant matrix transformation $^c\mathbf{T}_e$ is defined as

$$^c\mathbf{T}_e = \begin{bmatrix} 0 & 1 & 0 & 0 \\ -1 & 0 & 0 & 0.05 \\ 0 & 0 & 1 & 0.08 \\ 0 & 0 & 0 & 1 \end{bmatrix}. \tag{34}$$

Model and control parameters were set as: $\lambda = 1.5$, $\beta = 0.2$, $k_0 = 2.6$, $L = 0.2355$ m, $l = 0.15$ m, $m_2 = m_3 = 1.3$ kg and $m_4 + m_5 = 2$ kg. Mass $m_1$ is not required to compute the gravity vector. All parameters were selected experimentally to achieve the best performance.

## Simulation results

In simulation tests, four coordinates of interest points in the image were considered, where the desired image features $\mathbf{s}_d$ was set as

$$\mathbf{s}_d = \begin{bmatrix} -0.1302 & -0.1492 & 0.1302 & -0.1492 & 0.1302 & 0.1113 & -0.1302 & 0.1113 \end{bmatrix}^T \tag{35}$$

Moreover, the intrinsic parameters of the camera were set as $f = 600$, $C_u = 320$, $C_v = 240$. The simulation scenario of the ideal case is given in Fig. 3.

The ideal simulation aims to show that the second task in Eq. (26) is increasing the manipulability measure while performing the IBVS task with high priority.

Figure 4 shows the results when the secondary task is not included. Figure 4A the dashed lines represent the starting positions, while the solid lines indicate the desired positions. The colored dashed lines illustrate the trajectories of each keypoint. Although the keypoints reach the desired pose, Fig. 4B shows that, since the secondary task is not included, the manipulability measure does not exceed 0.3.

Conversely, in Fig. 5, we present the results of the same simulation, but with the secondary task included. It is evident that the manipulability measure increases, in contrast to the previous experiment (Fig. 4).

As we can see, Figs. 4A and 5A the image point trajectories (colored dashed lines) confirm that the primary task is being performed successfully.

In addition, in Fig. 6A we show the transition of the image features, where the dashed lines can be seen converging at the desired positions, represented by solid lines. In Fig. 6B we present the joint velocities of the mobile manipulator while performing both the

**Peer**J Computer Science

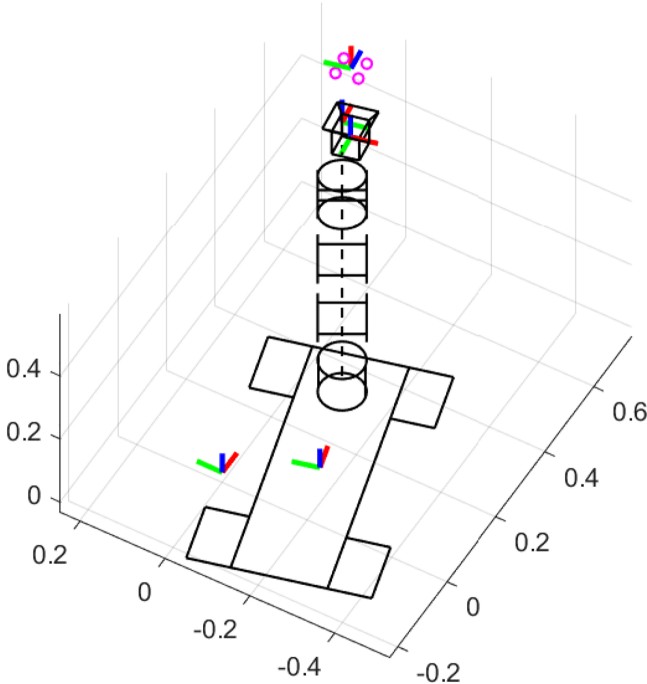

**Figure 3 Simulation scenario of the ideal case.**

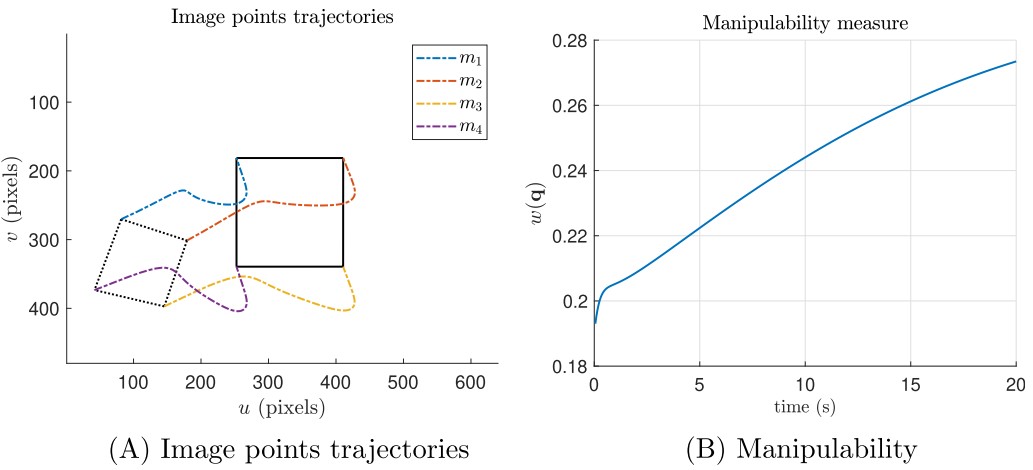

(A) Image points trajectories    (B) Manipulability

**Figure 4 Ideal simulations without the manipulability term (secondary task).** (A) Shows the trajectories of the features in pixel units and (B) shows the manipulability index.

primary and secondary tasks. It can be observed that the velocities converge smoothly to a neighborhood of zero, indicating that the desired positions have been reached.

A singularity test experiment is performed to validate the capacities of the proposed approach to avoid singularities and smooth out discontinuities by inverting the Jacobian matrix based on DLS, see Eq. (23). In this experiment, we also compared the proposal against the classical IBVS

$$\mathbf{v}_w = -\lambda (\mathbf{L_s}^c \mathbf{V}_w \mathbf{J}(\mathbf{q}))^+ \mathbf{e} \tag{36}$$

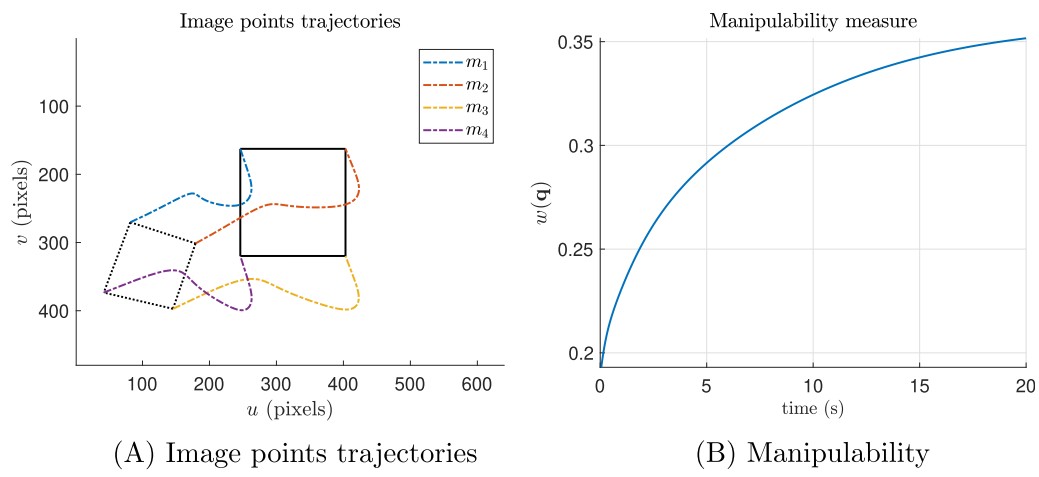

(A) Image points trajectories      (B) Manipulability

**Figure 5 Ideal simulations with the manipulability term (secondary task).** (A) The trajectories of the features in pixel units and (B) the manipulability index.

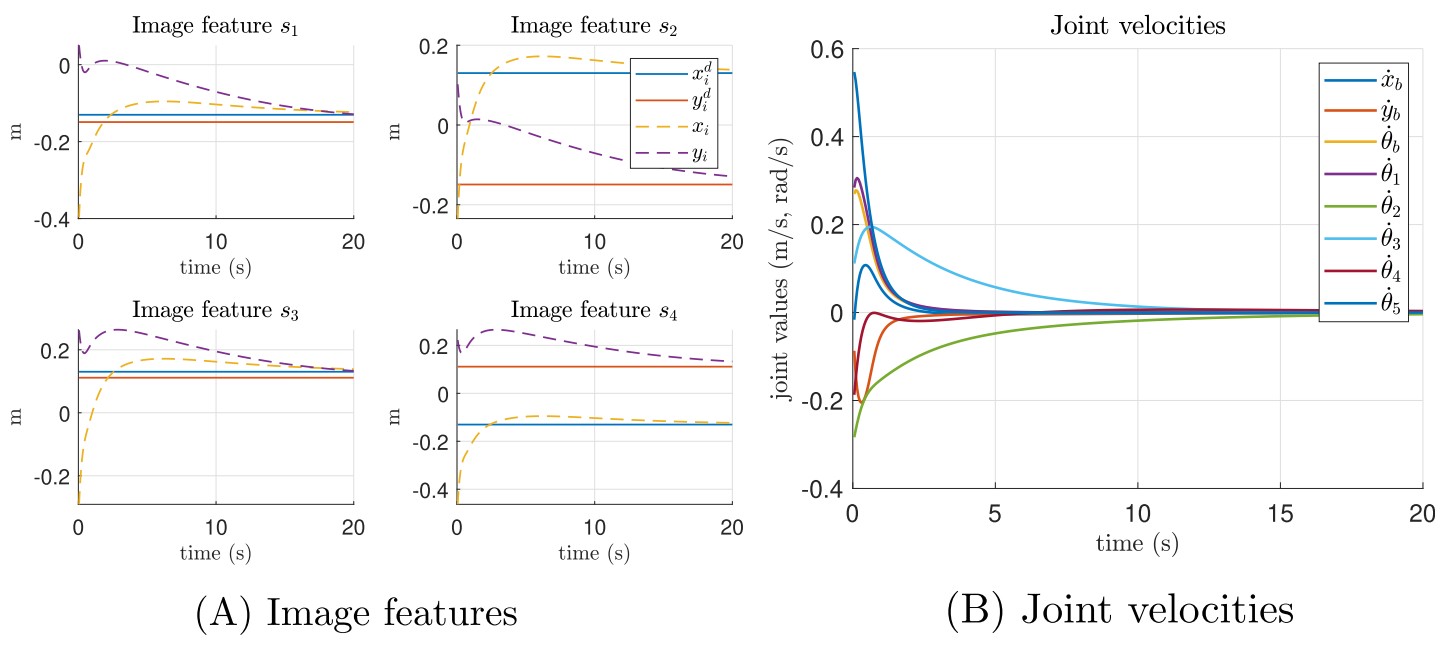

(A) Image features      (B) Joint velocities

**Figure 6 Ideal simulation.** (A) Image features and (B) joint velocities.

where the inversion of $\mathbf{A}^+$ with $\mathbf{A} = \mathbf{L_s}^c\mathbf{V}_w\mathbf{J}(\mathbf{q})$ was performed in two ways, using singular value decomposition (SVD) and using the well-known Moore-Penrose pseudoinverse $\mathbf{A}^+ = \mathbf{A}^T\left(\mathbf{A}\mathbf{A}^T\right)^{-1}$. SVD in the form $\mathbf{A} = \mathbf{U}\Sigma\mathbf{V}$ is another method to deal with singularities. To compute the inverse of $\mathbf{A}$, singular values along the diagonal of $\Sigma$ that are less or equal to a tolerance are treated as zero. Then, all the non-zero values of $\Sigma$ can be inverted.

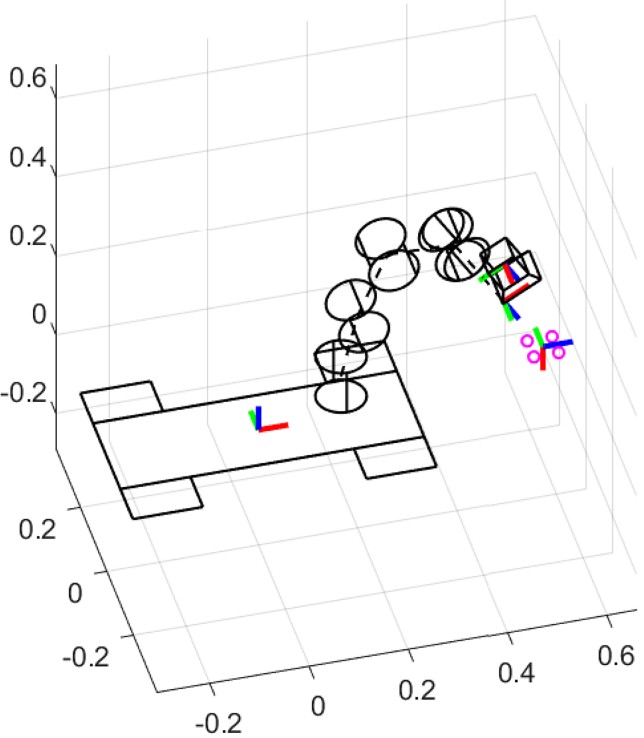

**Figure 7 Simulation scenario for the singularity test experiment.**

Figure 7 illustrates the simulation scenario to perform the singularity test experiment. To identify singularities, Monte Carlo experiments were performed to randomly select numerical joint configurations that provoke the determinant of $\mathbf{JJ}^T$ is close to 0. Then, four coordinates of interest points in the 3D world were intentionally placed near the position of the end-effector related to a singular joint configuration.

Figure 8 presents the joint positions results along time of the singularity test experiment. As can be seen in Fig. 8A, the Moore-Penrose pseudoinverse suffers from singularities around 9 s of simulation time. Abrupt changes in joint position are presented. Figure 8B shows that SVD reduces the impact of singularities and all joint positions are admissible. However, Fig. 8C demonstrates that the proposed approach significantly reduces the impact of singularities, since joint positions are not perturbed at all.

The joint velocity results of the singularity test experiment are reported in Fig. 9. In the case of the Moore-Penrose pseudoinverse, the presence of singularities provokes inadmissible joint velocities as shown in Fig. 9A. Despite the SVD reducing the impact of singularities, discontinuities are presented in some joint velocities which are not recommended for real-world applications, see Fig. 9B. Moreover, the joint velocities reported by the proposed approach are smooth, demonstrating that DLS smooth out discontinuities, see Fig. 9C. All velocities converge to a neighborhood of zero with admissible joint positions.

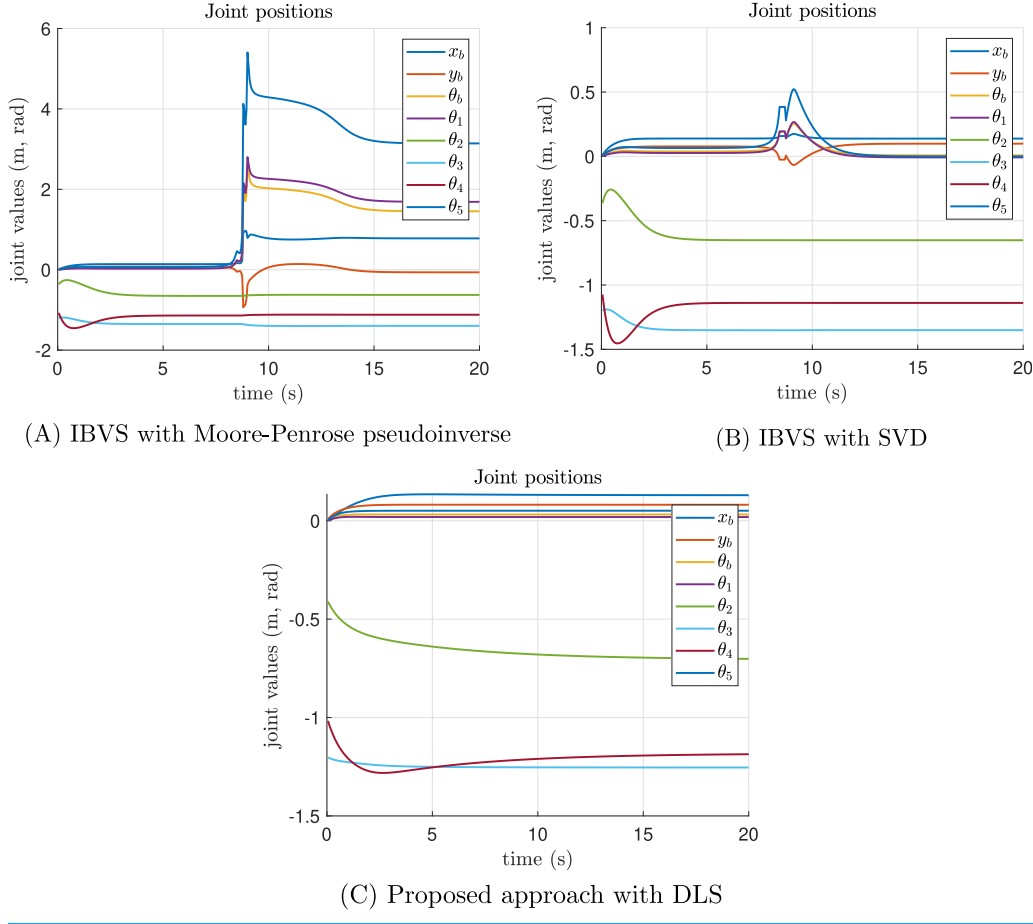

**Figure 8 Joint position results of the singularity test experiment.** (A) The joint position results of classical IBVS with Moore-Penrose pseudoinverse, (B) the classical IBVS with SVD to avoid singularities, and (C) the proposed IBVS scheme with DLS.

The image points trajectories of the singularity test experiment are provided in Fig. 10. As expected, Fig. 10A shows that the task of IVBS with Moore-Penrose pseudoinverse failed. In this case, the image points trajectories present abrupt changes that go out of the image boundary. Moreover, Fig. 10B reports that the IVBS with SVD reached the desired image features, despite the velocities discontinuities. Finally, Fig. 10C illustrates that the proposed approach with DLS successfully reached the desired image features.

Now, the proposed approach is tested in the Coppelia simulator. These simulations aim to test the proposed scheme under non-modeled dynamics and gravity compensation. The setup of these simulations is shown in Fig. 11.

From this experiment, it is important to report the error in the image features, as illustrated in Fig. 12. The figure clearly shows an offset in the $y$ direction, which can be attributed to the influence of gravity on the system. This offset indicates that the image features are not aligning perfectly with the expected positions, likely due to gravitational effects acting on the manipulator during the operation.

Figure 13 presents the results of implementing gravity compensation. The data clearly demonstrate that the error in the $y$ direction has been significantly reduced as a result of

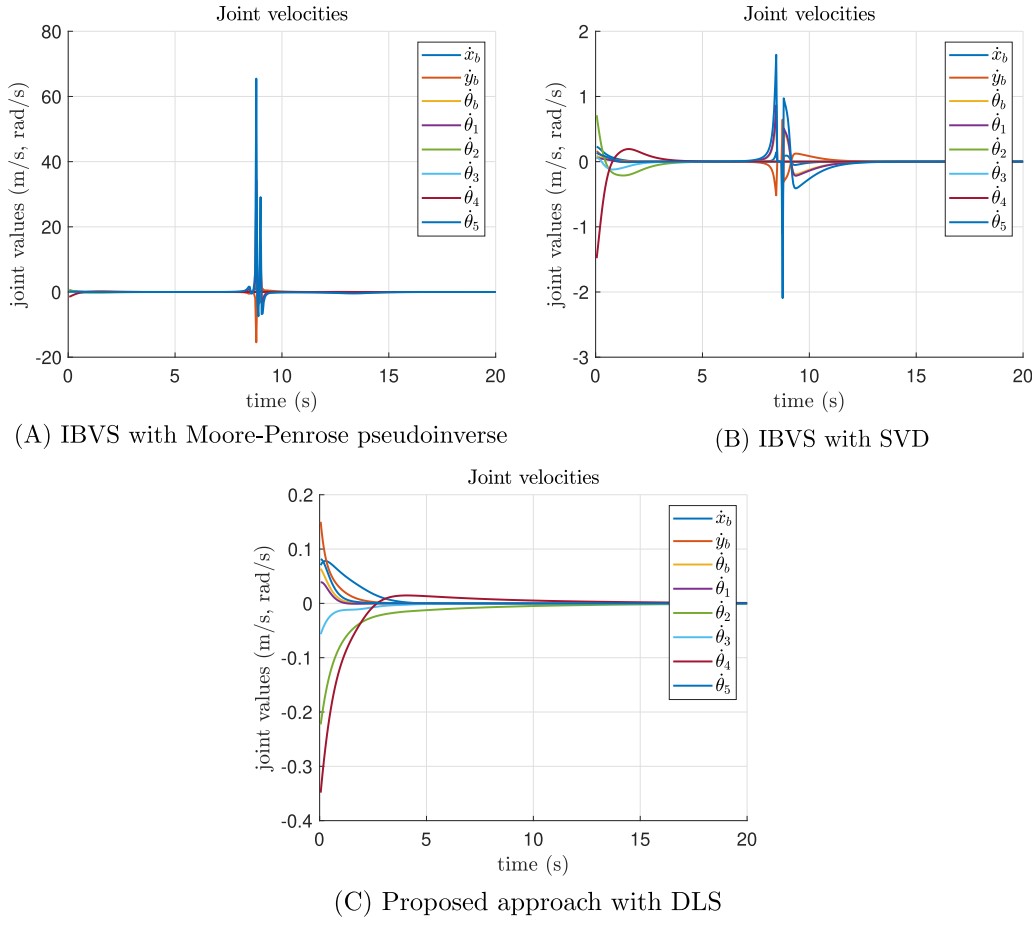

**Figure 9 Joint velocity results of the singularity test experiment.** (A) The joint velocity results of classical IBVS with Moore-Penrose pseudoinverse, (B) the classical IBVS with SVD to avoid singularities, and (C) the proposed IBVS scheme with DLS.

this compensation technique. This improvement indicates that the gravity compensation has effectively minimized the offset previously observed (Fig. 12), leading to better alignment of the image features with their expected positions.

## Real-world experimental results

The experiments are conducted using the Kuka Youbot omnidirectional platform, which has an initial pose as shown in Fig. 14. The QR code is placed within the field of view of this initial position. The platform is equipped with encoders which have been used to measure joint positions. Moreover, Encoders have been also used for the wheel odometry to estimate the pose of the mobile platform.

Since the surface where the mobile platform moves is not completely flat, external disturbances by surface imperfections can produce slippages that cause errors in odometry. Even with the lack of accuracy in platform pose estimation, the proposed IBVS succeeds since it does not rely on the internal odometry (*Li & Xiong, 2021*; *Jo & Chwa, 2023*), but instead the visual feedback from the QR code.

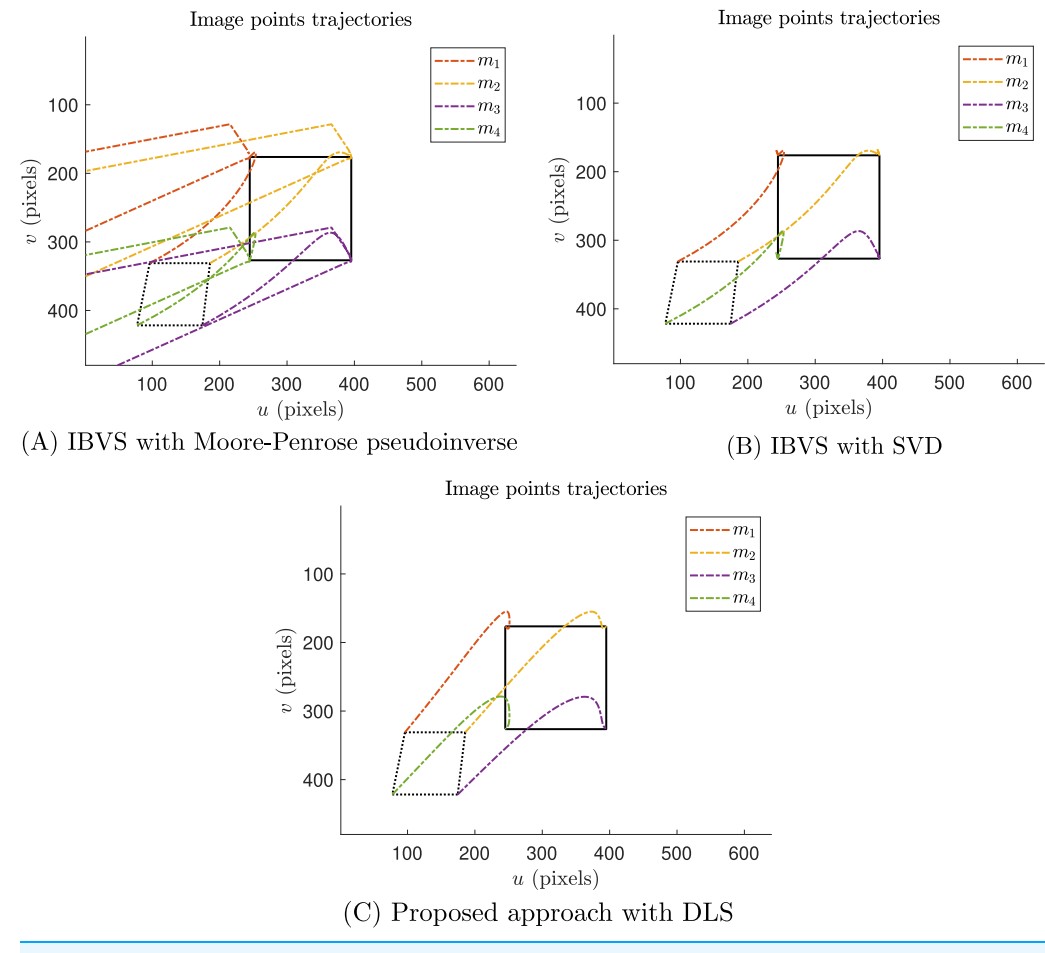

**Figure 10** **Image points trajectories of the singularity test experiment.** (A) Thee trajectories of the features of classical IBVS with Moore-Penrose pseudoinverse, (B) the classical IBVS with SVD to avoid singularities, and (C) the proposed IBVS scheme with DLS.

For this experiment, we measure four image features from a QR code. Figure 14 shows the starting and ending positions in an example of the experimental setup. This experiment was performed over 30 s. In this scenario, the desired image features $\mathbf{s}_d$ was set as

$$\mathbf{s}_d = \begin{bmatrix} 0.1459 & -0.17507 & -0.0865 & -0.1705 & -0.0929 & 0.0595 & 0.1491 & 0.0611 \end{bmatrix}^T \quad (37)$$

Moreover, the intrinsic parameters of the camera were calibrated as $f = 623.71$, $C_u = 317.97864$, $C_v = 257.93234$.

The joint velocities and positions from the experiment are presented in Fig. 15. Although the control signals are not as smooth as those observed in the simulations, Fig. 15A shows that the velocities converge to a neighborhood of zero. Several challenges arose during the real-world implementation, including unmodeled dynamics, external perturbations, robot wear, and surface imperfections. Despite these challenges, the joint positions are still achieved smoothly, as shown in Fig. 15B.

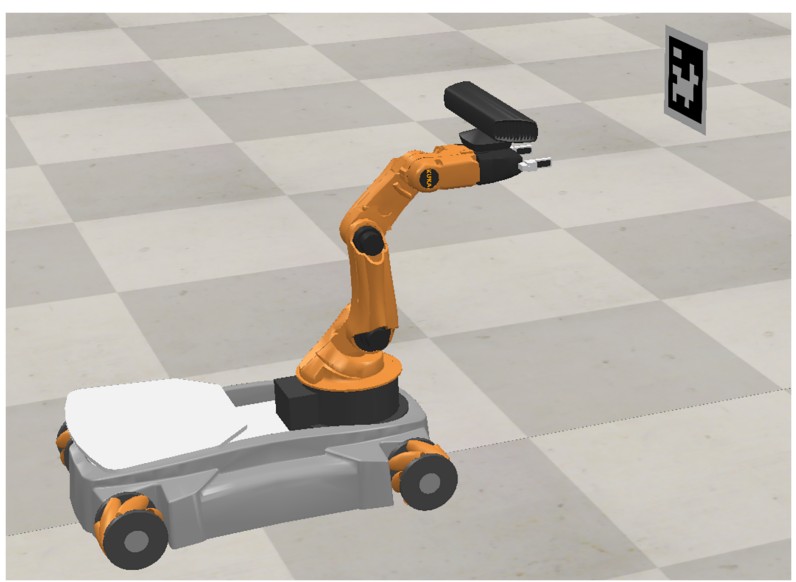

**Figure 11 Setup in Coppelia simulator.** Photo credit: Javier Gomez-Avila.

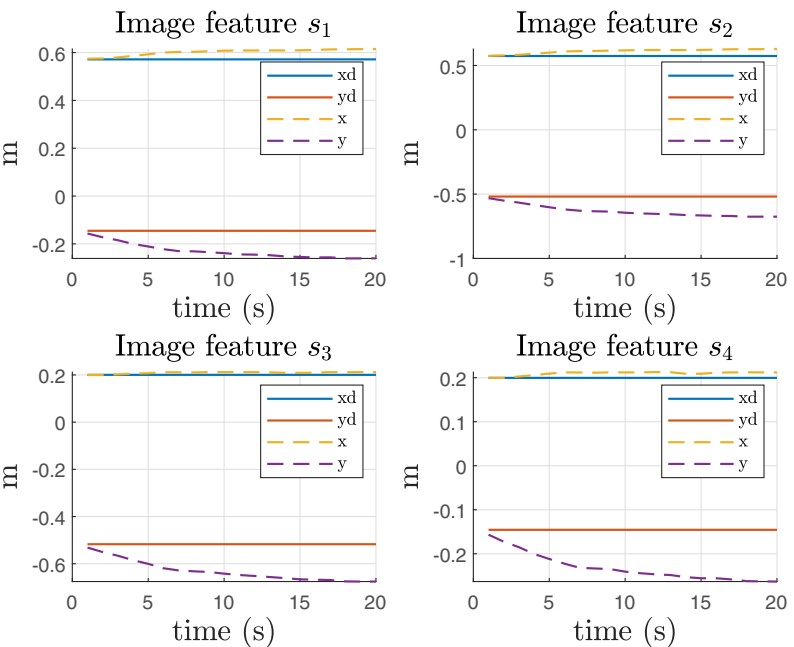

**Figure 12 Image features in Coppelia simulator before the gravity compensation.**

Figure 16A presents the results of the image feature transition and Fig. 16B image point trajectories. The reported results indicate that the reference position is achieved, even in the presence of noisy image features from QR detection. Additionally, Fig. 16A shows that the offset caused by gravity has been mitigated. In Fig. 16B, the black solid square represents the desired position, while the red square indicates the final position.

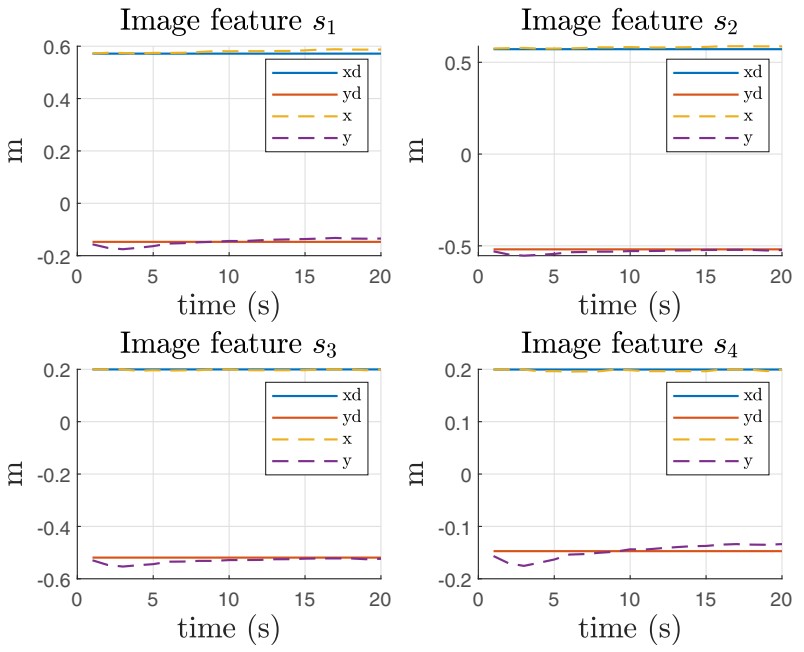

**Figure 13 Image features in Coppelia simulator with compensation of gravity.**

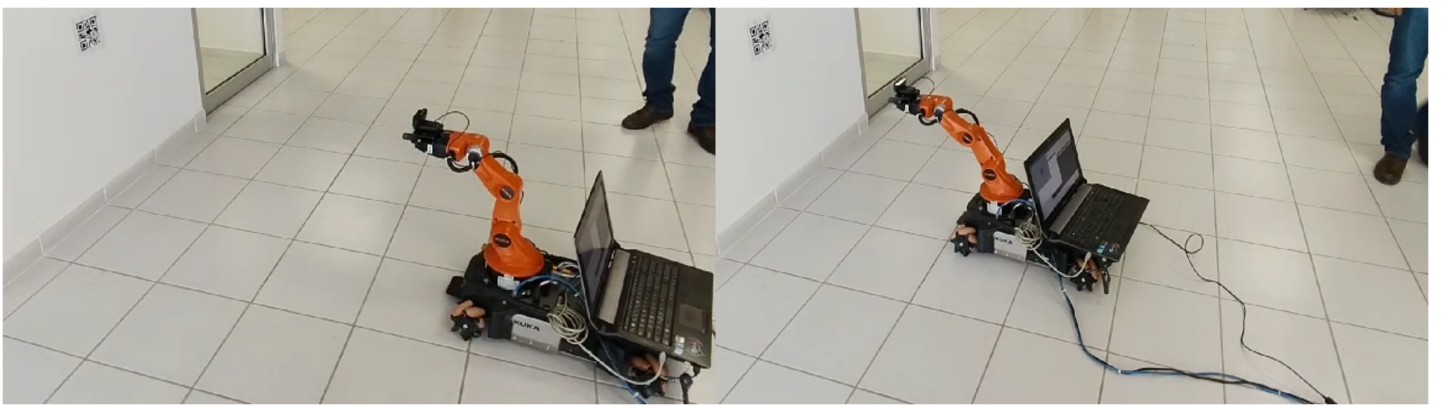

**Figure 14 Experimental setup. Initial (left image) and final (right image) positions.** Photo credit: Jesus Hernandez-Barragan.

Finally, Fig. 17 shows that the manipulability measure is increased as expected.

## DISCUSSION

Ideal simulations addressed the task prioritization scheme, where the eye-in-hand IBVS was the highest priority task, and manipulability maximization was the secondary task. Simulation results demonstrate that both tasks performed successfully with smooth image trajectories and control actions while maximizing the manipulability measure. Notice that with higher manipulability, it will reduce the incidence of singularities.

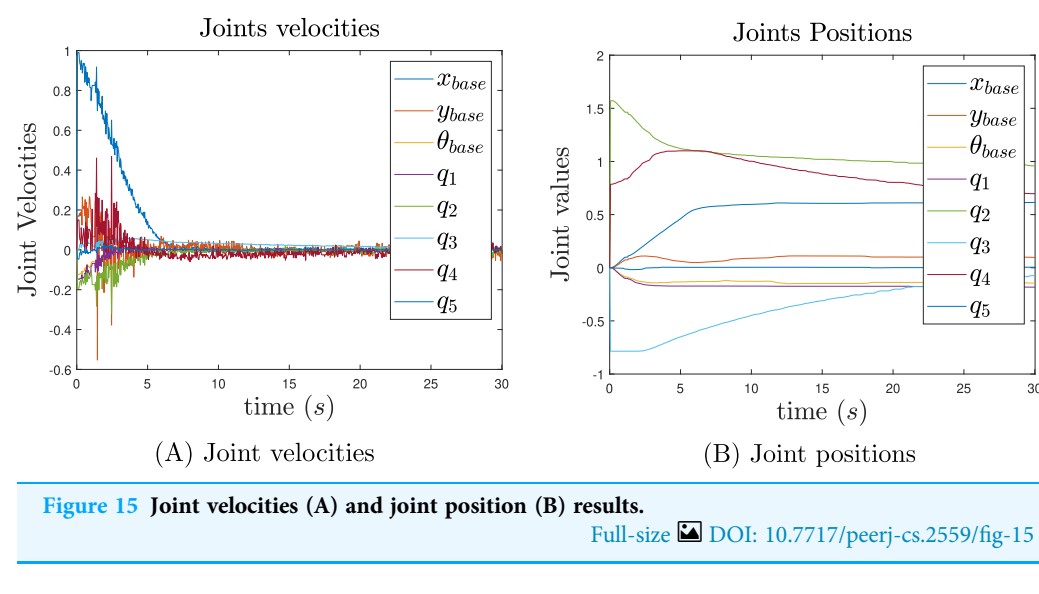

**Figure 15 Joint velocities (A) and joint position (B) results.**

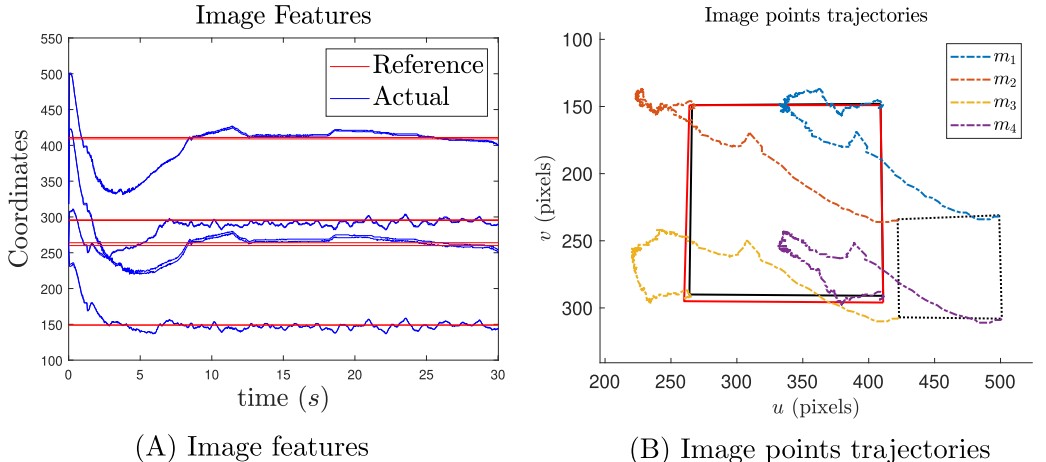

**Figure 16 Image feature results in pixel units (A) and their trajectories (B).**

A singularity test experiment was performed to demonstrate the singularity avoidance capacities of the proposed approach. In this case, the proposed approach was compared against the classical IBVS where the inversion of required matrices was performed with two methods, the Moore-Penrose pseudoinverse and SVD. Reported results demonstrate that IBVS with Moore-Penrose pseudoinverse performed poorly, the IBVS task practically failed since this method does not handle singularities. Moreover, IBVS with SVD reduces the impact of singularities, and the IBVS task succeeds. However, the presence of singularities provokes significant joint velocities discontinuities that are impractical in real-world implementations. Finally, the proposed approach demonstrates that avoids singularity configurations since joint position and joint velocities results are admissible and it smooths out discontinuities.

Simulations in the Coppelia simulator addressed the importance of gravity compensation. Simulation results without the compensation term report a significant error

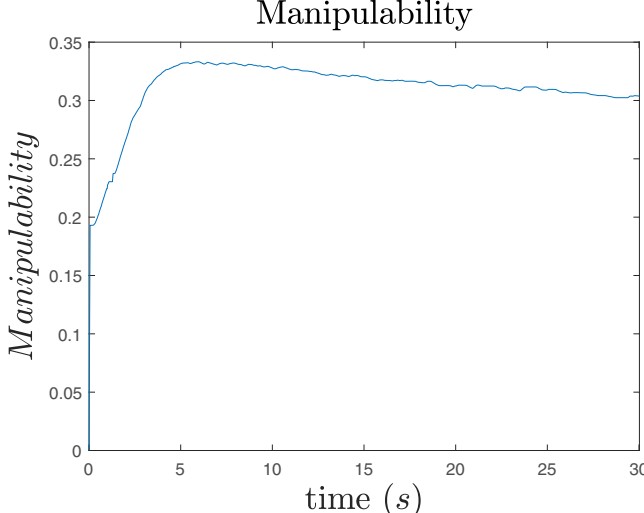

**Figure 17 Manipulability.**

in the y-axis of each image feature. However, the gravity compensation term effectively reduced this inconvenience.

The real-world experiments with the 8 DOF KUKA Youbot address the applicability of the proposed approach. The eye-in-hand IBVS was performed successfully with manipulability maximization along with gravity compensation. Although the input control signals of the experiments were not as smooth as in the simulations, position joint trajectories are still smooth. Notice that control action tries to compensate for non-modeled dynamics, external perturbations, and other drawbacks of the physical implementations which are not considered in simulated scenarios. Despite all these inconveniences, the proposed approach succeeds.

We use kinematic modeling for both the manipulator and the mobile platform to implement IBVS. No dynamic modeling is considered except for the gravity compensation.

Kinematic modeling uncertainties, such as kinematic modeling errors of the platform in Eq. (18) and the manipulator (Table 1) may arise since constants are defined based on technical specifications and practical procedures that have error tolerances. Additionally, there are parametric errors that can occur due to intrinsic and extrinsic camera calibration.

Finally, although we do not employ any techniques to address noisy image features (such as a Kalman filter) we still achieve the desired image features.

In this work, the VS task was implemented on an omnidirectional mobile manipulator. We let as future work, the design of kinematic modeling that includes mobile platforms with nonholonomic constraints. Moreover, the dynamic modeling of the mobile manipulator can be also included in the VS scheme. Finally, this work can be extended to the use of the eye-to-hand configuration.

## CONCLUSIONS

The article presented satisfactory results from implementing eye-in-hand IBVS enhanced with manipulability maximization and gravity compensation. A redundant

omnidirectional mobile manipulator KUKA Youbot was considered as a case study. Both, simulation and real-time experimentation demonstrated the effectiveness of this approach.

To demonstrate the effectiveness of our proposal and address the challenge of inverting $\mathbf{J(q)}^+$ near singularities, we applied the damped least-squares (DLS) inverse method. In simulation, the mobile manipulator was directed to approach a kinematic singularity. Results show that, with the classical method, velocities escalate significantly as the manipulator moves toward this configuration. In contrast, our proposed approach enables the manipulator to avoid the singular configuration, consistently maintaining velocities within acceptable limits.

Additionally, the secondary task, aimed at increasing the manipulability index, assists in ensuring that once the primary task is completed (reducing the positional error of the QR code in the image), the manipulator moves away from singular configurations. This approach not only helps achieve the primary positioning goal but also enhances the stability and performance of the system by avoiding configurations where control may become unstable.

### Funding
This research was funded by the University of Guadalajara through "Programa de Fortalecimiento de Institutos, Centro y Laboratorios de Investigación 2024". The funders had no role in study design, data collection and analysis, decision to publish, or preparation of the manuscript.

### Grant Disclosures
The following grant information was disclosed by the authors:
University of Guadalajara through "Programa de Fortalecimiento de Institutos, Centro y Laboratorios de Investigación 2024".

### Competing Interests
The authors declare that they have no competing interests.

### Author Contributions
- Jesus Hernandez-Barragan performed the experiments, performed the computation work, prepared figures and/or tables, and approved the final draft.
- Carlos Villaseñor conceived and designed the experiments, performed the computation work, authored or reviewed drafts of the article, and approved the final draft.
- Carlos Lopez-Franco analyzed the data, authored or reviewed drafts of the article, and approved the final draft.
- Nancy Arana-Daniel conceived and designed the experiments, analyzed the data, authored or reviewed drafts of the article, and approved the final draft.
- Javier Gomez-Avila performed the experiments, prepared figures and/or tables, and approved the final draft.

## Data Availability

Raw data and code are available at GitHub and Zenodo:

- https://github.com/Jegovila/IBVS_YouBot.

- Dr. Javier Gómez. (2024). Jegovila/IBVS_YouBot: v1.0.0 (v1.0.0). Zenodo. https://doi.org/10.5281/zenodo.14035156.

## Supplemental Information

Supplemental information for this article can be found online at http://dx.doi.org/10.7717/peerj-cs.2559#supplemental-information.

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
