# Peer review of "Image based visual servoing with kinematic singularity avoidance for mobile manipulator"

_PeerJ Computer Science, doi:10.7717/peerj-cs.2559_

## Round 0.1 · original submission · Major Revisions

Dear authors,
You are advised to critically respond to all comments point by point when preparing an updated version of the manuscript and while preparing for the rebuttal letter. Please address all comments/suggestions provided by reviewers, considering whether they should be added to the new version of the manuscript.

Kind regards,
PCoelho

Reviewer 1 ·

Basic reporting

References cited by the author are for listing purposes only. Please describe in more detail the Visual servoing techniques that have been applied. The advantages and disadvantages of existing methods. From there, present the differences and contributions of this study compared to existing works, and highlight the novelty and importance of this study.

Experimental design

No comment

Validity of the findings

- The paper states that the proposed method can solve the kinematic singularities problem. However, in the experimental results, there is no case where the proposed method is shown to be effective in reducing singularities. Please perform more experiments to better describe this issue.
- Compare the obtained results with traditional VS methods to see the novelty of the proposed method.
- The experimental results are also not clearly described. Please describe the experimental results images in more detail instead of just repeating their titles in the paragraph (Fig. 4, to Fig. 12)
- The conclusion should be rewritten in more detail. Briefly describe how the proposed research method solved the original problem and the experimental results obtained.

Reviewer 2 ·

Basic reporting

The paper presents an implementation of Image-Based Visual Servoing (IBVS) for a redundant mobile manipulator, specifically using an eye-in-hand configuration. The main challenge addressed is the issue of kinematic singularities that arise when using the Jacobian matrix in conventional visual servoing schemes. To mitigate this, the authors propose using Damped Least Squares (DLS) to improve the inversion of the Jacobian matrix, thereby reducing singularities and smoothing out discontinuities. Moreover, a secondary task prioritization and scheduling method is proposed to handle tasks such as maximizing the manipulability of the manipulator and avoiding singularities.

Overall, the paper is reasonably well-written. However, there are several major points that should be addressed:

1. Literature Review: The literature review is not well-organized and incomplete. Several references are unrelated to the main objective of the paper, particularly in the first and second paragraphs of the introduction. This weakens the overall quality of the paper. The authors should reorganize the introduction to focus more on IBVS and mobile manipulator systems.

2. Missing Relevant Works: Many recent and highly relevant works on IBVS in mobile manipulator systems have been overlooked. Most of the references cited are outdated, and some are not relevant. I recommend the authors consider the following highly related and recent works and their references to broaden and complete the literature review:
- Control barrier function-based visual servoing for Mobile Manipulator Systems under functional limitations, *Elsevier Robotics and Autonomous Systems*, DOI: https://doi.org/10.1016/j.robot.2024.104813.
- A hybrid visual servo control method for simultaneously controlling a nonholonomic mobile and a manipulator, *Frontiers of Information Technology & Electronic Engineering*, 2021.
- Image-based visual servoing control of robot manipulators using a hybrid algorithm with feature constraints, DOI: 10.1109/ACCESS.2020.3042207
- Image-based visual servoing for floating base mobile manipulator systems with prescribed performance under operational constraints, DOI: https://doi.org/10.3390/machines10070547.
- Robust Hybrid Visual Servoing of Omnidirectional Mobile Manipulator With Kinematic Uncertainties Using a Single Camera, DOI: 10.1109/TCYB.2023.3238820
- Prescribed performance image-based visual servoing under field of view constraints, DOI: 10.1109/IROS.2014.6942934
- Adaptive Image-based Visual Servoing of Mobile Manipulator with an Uncalibrated Fixed Camera, DOI: 10.1109/RCAR49640.2020.9303296

3. Technical Corrections:
- Line 86: The term *m(t)* is incorrectly referred to as the coordinates of interest points. The vector *s* should represent the coordinates of the interest points in the image plane, right?. The authors should correct this error, as *m(t)* might refer to something else, such as the configuration of the camera with respect to the object.
- Line 88: The notation *m(u,v)* for pixel coordinates is not properly introduced. The variable *m(u,v)* only appears in equation 1. Additionally, this explanation should be placed after equation 1 for better clarity. The authors should improve this section to enhance the paper's readability.

4. Joint Limits in Null Space Technique: The null space technique discussed could also be applied to avoid arm joint limits. It would be helpful to provide a remark stating that the null space techniques can be used to satisfy manipulator joint limits. I recommend adding this remark after equation 26.

5. Real-world experiments : I would like to ask the authors to make a remark discussing the real-world implementation of this work. In particular, there is a short discussion after equation 34, however, it would be extended by saying what sensors you need, and also discussing the effect of uncertainties and unmodeled dynamics.

Experimental design

The concepts of unmodeled dynamics, uncertainties, and noise could be addressed through a robust stability analysis. However, this is not a requirement from the reviewer. The only related suggestion, as already mentioned in point 5, is that a remark discussing the real-world implementation of this work, specifically considering noise and uncertainties, should be provided.

Validity of the findings

no comment

Additional comments

The article provides a well-structured approach to addressing kinematic singularities in visual servoing for mobile manipulators using the Damped Least Squares method. However, the paper could benefit from further validation through additional experiments, compared with other singularity avoidance techniques. Incorporating these aspects would strengthen the validity of the findings but it is NOT required by the reviewer.

---

## Round 0.2 · accepted · Accept

Dear authors, we are pleased to verify that you meet the reviewer's valuable feedback to improve your research.

Thank you for considering PeerJ Computer Science and submitting your work.

Reviewer 1 ·

Basic reporting

The author's explanations are clear, satisfactory and have helped clarify any unclear points. Edits and additions have been made appropriately.

Experimental design

no comment

Validity of the findings

no comment

Additional comments

no comment

Reviewer 2 ·

Basic reporting

After carefully reviewing the authors' responses and cross-referencing them with the material in the article, I believe the authors have sufficiently addressed my concerns and questions. I would like to thank the authors for considering my comments carefully. Furthermore, I appreciate the additional experiments conducted and the more thorough explanations provided for the results. I have no further questions and believe the paper is now suitable for consideration for publication.

Experimental design

I appreciate the additional experiments conducted

Validity of the findings

...

Additional comments

...